# In Vivo Reflectance Confocal Microscopy Applied to Acral Melanocytic Lesions: A Systematic Review of the Literature

**DOI:** 10.3390/diagnostics14192134

**Published:** 2024-09-25

**Authors:** Camilla Chello, Simone Cappilli, Luca Pellegrino, Simone Michelini, Gerardo Palmisano, Giuseppe Gemma, Marisa Salvi, Carmen Cantisani, Alessandro Di Stefani, Ketty Peris, Giovanni Pellacani

**Affiliations:** 1Dermatology Clinic, Department of Clinical Internal, Anesthesiological and Cardiovascular Sciences, Sapienza University of Rome, 00185 Rome, Italy; camilla.chello@gmail.com (C.C.); simo.mik@hotmail.it (S.M.); giuseppe.gemma98@gmail.com (G.G.); carmen.cantisani@uniroma1.it (C.C.); pellacani.giovanni@gmail.com (G.P.); 2Dermatologia, Dipartimento di Medicina e Chirurgia Traslazionale, Università Cattolica del Sacro Cuore, 00168 Rome, Italy; luca.pellegrino777@gmail.com (L.P.); palmisanogerardo@gmail.com (G.P.); marisa.salvi94@gmail.com (M.S.); alessandro.distefani@gmail.com (A.D.S.); ketty.peris@unicatt.it (K.P.); 3UOC di Dermatologia, Dipartimento di Scienze Mediche e Chirurgiche, Fondazione Policlinico Universitario A. Gemelli—IRCCS, 00168 Rome, Italy

**Keywords:** acral lentiginous melanoma, reflectance confocal microscopy, clinical diagnosis

## Abstract

**Background**: Acral melanocytic lesions often pose a diagnostic and therapeutic challenge for many clinicians. Reflectance confocal microscopy (RCM) is an imaging technique widely used for the assessment of skin cancers. The aim of this review is to explore the applicability of RCM for the diagnosis of nevi and melanoma on the acral sites. **Methods**: Study selection was conducted based on the application of RCM for acral melanocytic lesions. All types of articles (original articles, short reports, and single case reports) were included in the analysis following PRISMA updated guidelines. **Results**: The search retrieved 18 papers according to the selection criteria; after removing duplicate records and additional articles by one or more of the exclusion criteria, a total of seven studies were carefully evaluated. **Conclusions**: RCM seems a valuable and useful additional tool for the diagnosis of acral melanocytic lesions, and its use may decrease the need for invasive procedures to some extent. Visualization of deeper layers may be achieved through mechanical removal of the superficial stratum corneum.

## 1. Introduction

Acral lentiginous melanoma (ALM) was firstly described by Arrington et al. in 1977 as a distinct subtype of melanoma involving acral skin of palms and soles [1]. Common risk factors for all subtypes of cutaneous melanoma (i.e., sun exposure, fair skin type, family or personal history of melanoma, pre-existing atypical melanocytic nevi), are not relevant to the development of ALM [2]. This variant is rare in Caucasians (1–7%), with a higher incidence in non-White individuals, accounting for up to 58% of all cutaneous melanomas in Asians and even more (60–70%) in Non-Hispanic Blacks [3]. Because of its frequently atypical clinical morphology, ALM is frequently misdiagnosed and may receive prolonged courses of inadequate therapy [2,3]. It is a biologically aggressive melanoma subtype and is thought to carry a worse prognosis when compared with other melanoma subtypes [3]. Clinically, it presents as a growing pigmented macule or patch with uneven margins and irregular pigmentation that, over the natural course of the disease, can develop a plaque or nodule, and occasionally ulcerate, as result of vertical growth [1,2,4]. Also, nail melanoma belongs to the pathology subtype of ALM. It is relatively rare, occurring almost always in adults and, more frequently, involving the first toe or thumb. As it develops in the nail matrix, what is macroscopically visible is a pigmented band on the nail plate (melanonychia) with an irregular pigmentation, originating from the proximal nail fold and extending towards the distal end of the nail plate [4,5]. A periungual pigmentation (Hutchinson sign) may be present in support of the clinical diagnosis of ALM [4,5]. Age, breslow thickness, and ulceration are poor prognostic factors for ALM, and clinical outcomes rely on an early diagnosis [3,4]. The utility of the standard ABCDE criteria (i.e., asymmetric shape, border, color, diameter, evolution) for its assessment has been questioned due to different presentation compared with other melanoma subtypes [6,7]. Hence, an alternative acronym CUBED (colored, uncertain, bleeding, enlarged, and delay) was proposed to facilitate clinical diagnosis; however, this acronym lacks specific morphologic criteria and was not evaluated in clinical setting [7]. Dermoscopy has been widely used as an adjunctive clinical tool for the assessment of melanocytic and non-melanocytic acral skin lesions, and its application has been demonstrated to increase sensitivity and specificity for the diagnosis of ALM, by allowing the visualization of diagnostic clues not visible with naked eye [4,5]. Indeed, ALM shows specific dermoscopic patterns, including the parallel ridge pattern (PRP) and irregular diffuse pigmentation, in contrast to benign acral melanocytic nevi commonly identified by one of the following major patterns: the parallel furrow pattern, the fibrillar pattern, or the lattice-like pattern [4,5,8] (Figure 1 and Figure 2). With dermoscopy, a prediction of the tumor thickness (in situ versus invasive ALM) may be supposed before histopathologic exam, since in situ ALM reveals specific colors (blue and white) and patterns (atypical vascular patterns, blue-white veil, and ulcers) with a reduced frequency rather than invasive form [9]. Two different dermoscopic algorithms, the three-step algorithm and the BRAAFF algorithm, have been proposed for a more accurate management of acquired melanocytic lesions on the acral sites [10,11]. The three-step algorithm was introduced by Saida and Koga in 2007 and considered the presence of the parallel ridge pattern and diameter of skin lesion, to differentiate early ALM and acral nevi. Based on this algorithm, skin lesions showing the parallel ridge pattern or measuring a diameter >7 mm without the typical benign pattern, should be confirmed by histopathology; all other lesions may be clinically followed over time [10]. A revised version of the algorithm in 2011 suggested to remove clinical follow-up for acral nevi with benign dermoscopic findings, as ALM mostly arises de novo [12]. In a single retrospective chart review, the three-step algorithm was found to have a sensitivity of 80.0%, specificity of 87.8%, positive predictive value of 44.4%, and negative predictive value of 97.2% for the diagnosis of ALM [13]. Considering the low sensitivity of the parallel ridge pattern in ALM in their case series, Lallas et al. proposed the BRAAFF checklist, to enhance the diagnosis of ALMs presenting an irregular blotch and the asymmetry of structures and colors [11]. The scoring system evaluates four positive patterns, irregular blotches (1 point), parallel ridge pattern (3 points), asymmetry of structures (1 point), and asymmetry of colors (1 point); and 2 negative features, parallel furrow pattern (−1 point) and fibrillar pattern (−1 point). Lesions with a total score of 1 or higher are suspicious for ALM and should be accurately investigated [11]. The parameters of sensitivity and specificity of the BRAAFF checklist were reported as 93.1% and 86.7%, respectively. [11] Although the use of dermatoscopy has improved the diagnostic accuracy for acral melanocytic lesions, a differential may be challenging, and a missed diagnosis may be fatal [9,10,11,12,13].

Reflectance confocal microscopy (RCM), also known as confocal laser scanning microscopy (CLSM), of the skin was first described in the early 1990s [14]. It is a widely used non-invasive in vivo skin imaging technique that allows the analysis of a wide variety of skin cancers and cutaneous inflammatory and infectious diseases [15,16]. Commonly available confocal microscopes use a low-power near-infrared laser beam (830-nm diode laser and power up to 35 mW), obtaining a horizontal viewing of the skin from the epidermis up to the dermis with real-time examination of skin lesions at a cellular-level resolution [15,16]. The laser light is a source of coherent, monochromatic light that penetrates the tissue and illuminates a single focal point. The different reflection index of cellular structures is captured and illustrated into a two-dimensional grey scale image by the software [15,16].

Two different devices with different designs for specific indications are available on the market. The Vivascope 1500 model provides a lateral resolution of about 1 μm and an axial resolution of 3–5 μm, reaching a depth of about 200–250 μm, thus allowing the visualization of the papillary dermis. In areas with the thinner epidermis, such as the face and mucous surfaces, it is possible to analyze the superficial reticular dermis as well. With this model, a metal ring with a plastic window is placed over the lesion and an immersion oil acts as an interface between the skin and the plastic ring [17,18,19]. Horizontal RCM scans of the skin produce multiple individual images of 0.5 × 0.5 mm^2^, forming a mosaic of up to 8 × 8 mm^2^. The Vivascope 3000 is a model with a portable probe easy to mobilize. The advantage of this model is the smaller dimension that allows access to hard-to-reach body areas, such as inguinal folds and between digits. The process to obtain images is faster with this device, although the images have a global small field of view of 1 × 1 mm^2^, with the ability to acquire vertical sections only (stacks) [17,18,20,21].

In the last decades, the application of RCM has resulted in a high diagnostic accuracy for the diagnosis of melanoma: a recent meta-analysis reported a sensitivity of 92% and a specificity of 70% for the detection of melanoma, with a higher diagnostic performance of RCM when compared with dermoscopy alone (sensitivity of 96% vs. 90% and specificity of 56% vs. 38% (95% CI, 0.34–0.42) [22]. In addition to its application for pigmented melanomas, RCM is also notable for the diagnosis of amelanotic/hypomelanotic melanomas, showing a higher sensitivity in comparison to dermoscopy (67% vs. 61%), with a similar specificity (89% vs. 90% [23]. If combined with dermoscopy, RCM reduces the number need to excise by 43.3% (from 5.3 to 3.0) in lesions suspicious for melanoma, reducing the rate of unnecessary benign excisions, and, thus, providing significant cost–benefit savings [24]. In a single-center study, 4320 unnecessary excisions were avoided in a year, saving over 280,000 EUR, thanks to its use [25]. In another investigation, 50.2% of biopsies were avoided by identifying benign lesions with RCM [26]. RCM utility is thought to be limited for acral pigmented and non-pigmented lesions due to the thick stratum corneum and decreased light penetration [15,16,17,18].

The aim of this review is to explore the applicability of RCM for the diagnosis of nevi and melanoma on the acral sites.

## 2. Materials and Methods

### 2.1. Search Strategy

We followed the Preferred Reporting Items for Systematic Reviews and Meta-Analyses (PRISMA) 2020 updated guidelines for reporting systematic reviews [27] (Figure 3). All types of publications (original article, case series, and case report) evaluating the application of RCM for acral melanocytic lesions were screened on 30 May 2024 in PubMed, Scopus, and Web of Science databases using the following Medical Subject Heading (MeSH) terms: “reflectance confocal microscopy” (or its acronym “RCM”) AND “acral melanocytic” OR “acral melanoma” OR “acral pigmented”.

### 2.2. Study Extraction and Synthesis

Two independent investigators (CC and GG) screened each study based on title and abstract. Titles and abstracts were screened to determine the eligibility for the full-text screening stage, and a reference list of selected articles was checked to see whether our previous search terms missed some articles. Screening conflicts were resolved by a third reviewer (SC). The following data were extracted for each study: author(s), year, design, and relevant findings (Table 1). One reviewer conducted data extraction (SC). Exclusion criteria were studies not in English language, narrative or systemic literature reviews, meta-analyses, and conference abstracts.

## 3. Results

We performed a search of articles with the MeSH terms previously mentioned. The search retrieved 18 papers according to the selection criteria; after removing duplicate records, and additional articles by one or more of the exclusion criteria, a total of seven studies were included in this review, as shown in the PRISMA flowchart summarizing the results of this process (Figure 3). A meta-analysis was not plausible due to low data. Instead, a narrative synthesis was conducted, synthesizing and grouping the results of the reviewed studies.

The first case of ALM assessed with RCM was reported by Kolm et al. in 2010 [28]. A 57-year-old man presented with a longstanding pigmented foot lesion showing rapid clinical changes. Dermoscopic examination revealed a parallel ridge pattern at the periphery, with an atypical network and blue-white veil in the central area. Further evaluation with RCM showed a widened honeycomb pattern in the granular layer and a high concentration of eccrine duct orifices with peri-glandular free melanin and bright cells that, with higher magnification, corresponded to pagetoid cells disposed in single unit and in clusters. As dermoscopy and confocal microscopy raised suspicious for melanoma, the skin lesion was surgical excised, and histopathology confirmed the diagnosis of ALM (Breslow 1 mm). Indeed, the pathologic specimen revealed an asymmetric melanocytic lesion with alternating nests, irregular in size and shape, and melanocytes aligned in solitary units at the junction. A pagetoid spread of melanocytes in the epidermis was focally prominent with cellular infiltration of the eccrine duct epithelium [28] (Table 1). RCM features of benign and malignant acral melanocytic lesions were then compared in a case series, including 17 ALM (6 hypomelanotic melanoma) and 26 acral nevi. All excised lesions showed atypical findings in dermoscopy, as ALMs showed pigmentation of the ridges (parallel ridge pattern and structureless pigmentation), and hypomelanotic melanomas were characterized by atypical vessels and milky red areas. The RCM analysis of the melanomas revealed, in the epidermis, an irregular honeycomb pattern (in situ melanoma) or a completely architectural disarray (invasive melanoma), due to the epidermal infiltration of atypical cells with different shapes (pagetoid, dendritic, and triangular) and granular dust-like bright particles as the main key clues. Atypical cells were also detected in the DEJ and upper dermis. The infiltration of the sweat duct structures by atypical bright cells represented an additional finding of malignant lesions. Nevi were defined by regular melanocytic nests at the DEJ and in the superficial dermis. In the RCM evaluation, it should be considered that benign melanocytic nevi could also show some pagetoid cells (with a monomorphic shape and low count) and melanocytic proliferation near glandular ducts, with a lower frequency and organized in nests [29].

RCM provided crucial hints also in the diagnosis of acral lesions mimicking ALM. A paper by Cheng et al. in 2016 described a rapidly growing pigmented lesion in a 31-year-old man with homogeneous brown pigmentation not preferentially involving furrows or ridges. As the lesion was rapidly enlarging, RCM was performed, revealing filamentous and bright superficial structures corresponding to fungal hyphae. Hence, the melanocytic nature of the lesion was excluded, and the diagnosis of tinea nigra was confirmed by skin scraping, avoiding biopsy, thus leading to its resolution with appropriate antifungal therapy [30]. Differently from the previous case, its application did not provide benefits in the in vivo evaluation of a pigmented papule on the right palmar hand in a 4-year-old girl. By dermoscopy, the authors observed changing worrying features, ultimately showing pigment network in the ridges and asymmetric dots and globules (crista-dotted pattern) at a 2-months follow-up visit. At this time RCM revealed atypia at the DEJ and intraepidermal melanocytic nests without cellular pleomorphism. Based on dermoscopic modifications and RCM superficial atypical cells, the lesion was excised, with histopathologic diagnosis of intraepidermal compound Spitz nevus, revealing spindled and epithelioid melanocytes at the junction along with epidermal pagetoid spreading of large, atypical melanocytes [31].

A greater visualization of deeper layers in acral lesions may be obtained through a superficial debriding of the stratum corneum. Two patients presented with equivocal lesions on the sole, showing dermoscopic characteristics of acral melanoma, including parallel ridge pattern, irregular globules, and diffuse hyperpigmentation. Confocal microscopy showed only a thick bright corneal layer and sweat ducts before debridement, while after such a procedure, a regular honeycomb pattern in the epidermis and dense dermal nests could be observed, suggesting the clinical diagnosis of benign melanocytic nevi [32]. In a single case of ALM previously treated with multiple surgical and topical interventions, RCM was performed to define peripheral margins of the tumor [33]. Scattered areas with pleomorphic bright structures (round, dendritic, and stellate), as corresponding to embedded melanocytic remnants discharged from the epidermis, were detected by RCM; subsequent punch biopsies obtained in the areas of concern confirmed the presence of residual melanoma [33].

Lastly, RCM features characterizing ALM were compared to histopathological findings, with the aim to determine the concordance between these two methods [34].

**Table 1 diagnostics-14-02134-t001:** Studies included in the review.

Author, Year of Publication	Type of Study	Number of Patients	RCM Analysis
Kolm et al., 2010 [28]	case report	1	correlation between dermoscopy, RCM and histology in ALM
Cinotti et al., 2016 [29]	short report	43 (17 ALM and 26 acral nevi)	determination of the RCM features of ALM and acral nevus
Cheng et al., 2016 [30]	case report	1	discrimination of an acral melanocytic lesion and tinea nigra
Iriarte et al., 2018 [31]	case report	1	RCM features of a pigmented Spitz nevus
John et al., 2019 [32]	case report	2	identification of typical benign melanocytic nevi after debridement of the stratum corneum
Natarelli et al., 2023 [33]	case report	1	analysis of peripheral margins of ALM to target the biopsy
Zou et al., 2024 [34]	original article	31	concordance between RCM features and histopathology results in ALM.

Legend: ALM, acral lentiginous melanoma; RCM, reflectance confocal microscopy.

RCM showed substantial concordance with histopathology in the diagnosis of ALM, as the results were consistent in 29 of 31 patients (93.5%) included in the study, with only two false positive RCM diagnosis. In an additional two cases, a diagnosis of melanoma was not made at initial histologic evaluation, even though a subsequent biopsy guided by RCM correctly diagnosed ALM. Hence, the sensitivity of RCM for the diagnosis of ALM was 100%, the specificity was 50%, the positive predictive value was 93.1%, and the negative predictive value was 100% [34]. The main findings characterizing ALM in RCM were as follows: epidermal disarray and disappearance of the honeycomb pattern; scattered spindled, dendritic cells and oval Paget cells; bright granules in the epidermis; disruption of the dermo-epidermal junction with disappearance of dermal papillae; and atypia grouped in nests or as a single unit. Based on these results, the authors considered the presence of epidermal dendritic cells in >30% of the skin lesion, and round and oval cells at the dermal-epidermal junction, as sensitive indicators for melanoma diagnosis. Bright granules were regarded as another clue for the diagnosis of malignancy [34] (Table 1).

## 4. Discussion

RCM is an imaging technology that provides noninvasive, in vivo evaluation of the skin at nearly histologic resolution [14,15,16,17]. Over the past two decades, its use in clinical dermatology has profoundly improved the performances of clinicians in the diagnosis of benign and malignant skin neoplasms, as well as for skin lesions of concern based on dermatoscopic findings [20,21,22,23,24]. According to the current literature focusing on the value of RCM for acral melanocytic lesions, RCM can provide sufficient key clues that raise suspicions of ALM [28,29,30,31,32,33,34]. Indeed, the presence of epidermal architectural disarray with unevenly distributed bright granular particles and cells; as well as alteration of the DEJ with loss of the dermal papillae, replaced by atypical cells in single units or nests; represent the main findings indicating an ALM and are similar to the classical criteria described in melanoma on other skin body areas [22,23,28,29,30,31,32,33,34] (Table 2). Acral melanocytic nevi are otherwise defined by a honeycomb pattern alternating with dark bands and regular globules, corresponding, respectively, to ridges and furrows and melanocytic nests on histopatology [29,31,32,34].

Hence, the current data in the literature suggest that RCM has the ability to analyze melanocytic lesions in acral body sites, as ALM shows characteristic features, revealed by RCM, related to the upward migration of atypical melanocytes in the superficial granular layer or stratum corneum of the epidermis. Such pathological changes may be observed in acral sites for the thinning of the epidermis occurring in ALM [34].

In the assessment of acral pigmented lesions, local trauma and superficial fungal infections (Tinea nigra) should also be considered in the differential diagnosis [13,30,35,36,37]. A fungal infection on the acral sites shows multiple highly refractile, branching, and round structures at the stratum corneum, related to the filamentous, septate hyphae, and arthroconidia of the fungus seen in microbiological tests and histopathology [35,36,37]. The detection of such features and the lack of RCM features indicative of a melanocytic nature of the lesion immediately allow a fast differential diagnosis between these two entities [29,35,36,37].

One of the main technical limitations is the loss resolution of RCM below 250 µm in depth, which may influence the evaluation of skin lesions in acral sites. Such a bias may be less relevant in the fingers or toes, where the skin is thinner compared to other acral sites, and after mechanically removing the most superficial layer of the skin lesion examined [32]. Indeed, deeper acquisitions can be obtained through the debridement of the stratum before performing an RCM exam, increasing the penetration depth of images with a high-resolution quality image [32]. In addition, the limited imaging depth can be overcome by integrating RCM with imaging techniques providing a higher pentetration depth, such as optical coherence tomograpy (OCT), which is able to capture skin structures up to 1.5 mm [38]. The combination of both RCM and OCT technical skills in a single medical device may represent the technique of choice for the assessment of acral skin lesions. A novel imaging device, line-field optical coherence tomography (LC-OCT) was introduced in 2018, as it combines principles of both RCM and OCT, acquiring in vivo images with cellular-level resolution in a non-invasive manner, showing similar patterns to histopatological slides. Indeed, it is capable of providing vertical section images up to superficial-mid dermis (~500 µm), horizontal section images, and three-dimensional (3D) cubes with a high definition (≅1.3-µm lateral resolution and ≅1.1-µm axial resolution) [39,40,41]. Since it has been shown that LC-OCT can provide diagnostic clues in skin diseases of different natures, including neoplastic, inflammatory, and infectious skin diseases, it gained popularity in the clinical setting [42,43,44,45,46,47,48,49,50]. The performance of this device to assess melanoma versus nevi seems to be high, and in a recent meta-analysis, high sensitivity (pooled 87%) and specificity (pooled 91%) were obtained by evaluating various skin tumors, including malignant melanoma [42,44,49]. Relying on its deeper depth imaging penetration, future studies are warranted to explore its value for melanocytic lesions on the palms and soles as, currently, this device has not been already employed to evaluate acral sites.

Another significant limitation is the scarce literature describing RCM application on acral melanocytic lesions; it mostly relies on case report/series and few original research studies. Indeed, only two research studies investigated RCM structures/criteria for ALM, and one of them made a comparison with melanocytic nevi. In addition, some melanomas were clinically evident in these studies, presenting as ulcerated, crusted, bleeding, and nodular, or extending to the nail fold [29,34]. The low data and the heterogeneous investigations explain why meta-analyses could not be conducted. Case reports and case series were, however, included as they provide relevant details that can contribute to tips in clinical practice. English-language restricting criterion may have represented a methodological bias.

## 5. Conclusions

In conclusion, RCM seems a valuable and useful additional tool for the diagnosis of acral melanocytic lesions, and its use may decrease the need for histopathological biopsy to some extent. However, monitoring acral skin lesions using RCM may be challenging, mainly for the technical limitation of depth acquisition, which may be potentially overcome in the near future by integrating new digital imaging with higher performance. As a general rule, when RCM features are equivocal and influence the diagnostic confidence of physicians, as ALM cannot be excluded, surgical excision should be safely performed.

## Figures and Tables

**Figure 1 diagnostics-14-02134-f001:**
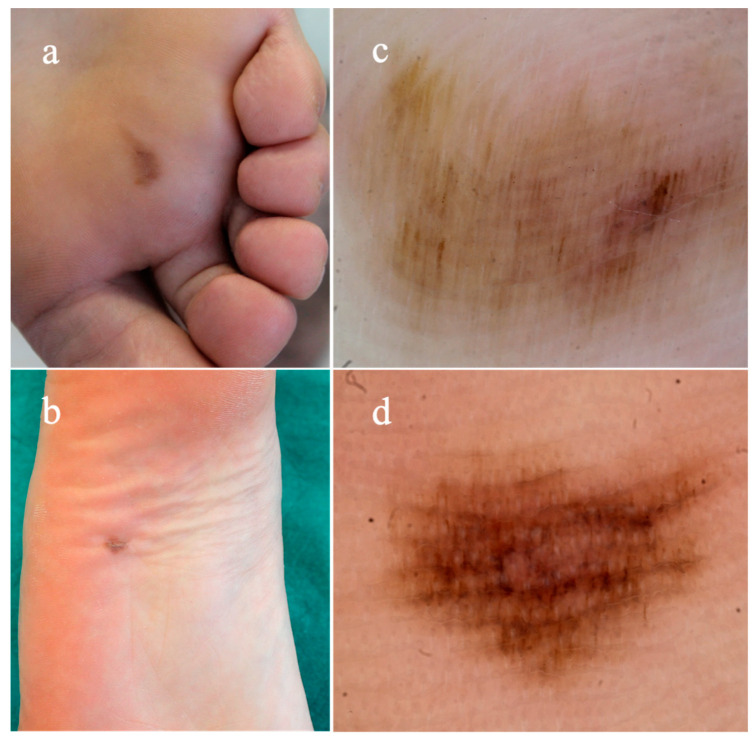
(**a**,**b**) Acral melanocytic nevi on the plantar surface showing in dermatoscopy, (**c**) parallel furrow pattern, and (**d**) fibrillar pattern.

**Figure 2 diagnostics-14-02134-f002:**
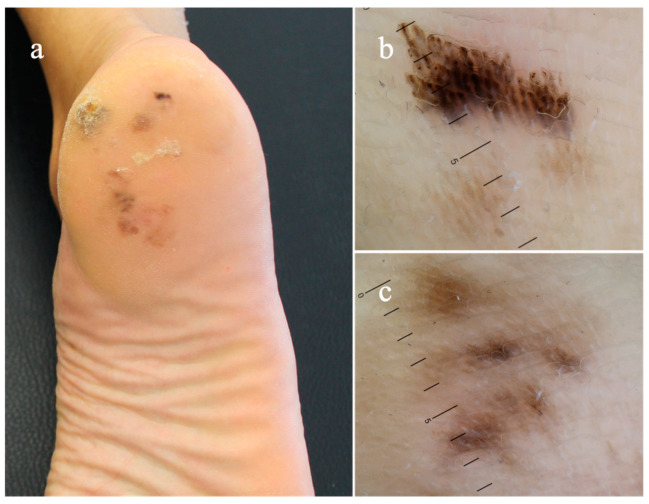
(**a**) Acral lentiginous melanoma revealing dermatoscopic features of (**b**) parallel ridge pattern and (**c**) asymmetry of structures and colors.

**Figure 3 diagnostics-14-02134-f003:**
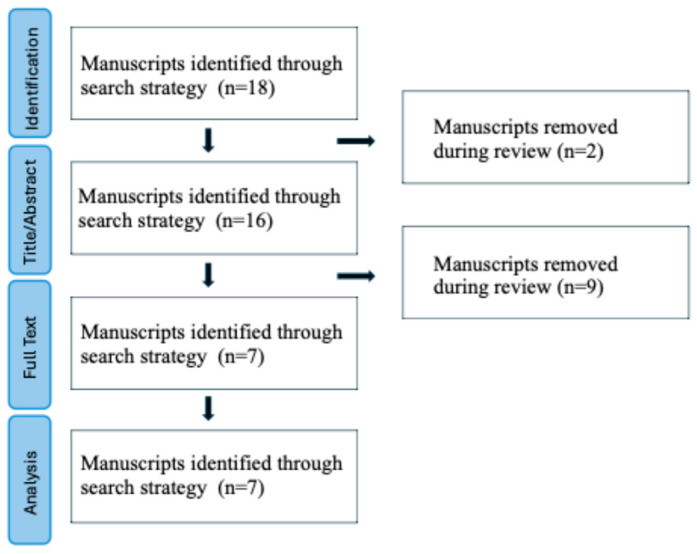
PRISMA flowchart diagram (adapted from Page et al., 2021 [27]).

**Table 2 diagnostics-14-02134-t002:** Confocal criteria of acral lentiginous melanoma.

	RCM
Epidermis	atypical honeycomb pattern/disarranged pattern pagetoid cells (roundish, dendritic, triangular) granular dust-like bright particles (free pigment)
DEJ and upper dermis	altered DEJ atypical bright cells arranged in sheets or nests

Legend: RCM, reflectance confocal microscopy; DEJ, dermal-epidermal junction.

## Data Availability

The data presented in this study are available on request from the corresponding author.

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
