# Peer review of "In Vivo Reflectance Confocal Microscopy Applied to Acral Melanocytic Lesions: A Systematic Review of the Literature"

_diagnostics, 2024, doi:10.3390/diagnostics14192134_

Round 1

Reviewer 1 Report

Comments and Suggestions for Authors

The authors provided a systematic review of the literature describing the application of RCM in diagnosing acral melanocytic lesions. Due to the lack of adequately designed studies in this field, only two retrospective and observational studies evaluated RCM structures/criteria for ALM. Still, only one of them included the comparator group of melanocytic nevi. Including the two case reports, this review describes 50 acral melanomas diagnosed with RCM.

The detailed analysis of the cited studies provided insights into the much more limited utility of the RCM in diagnosing ALM, as most cases included advanced and clinically evident melanomas, which wasn't pointed out in this article.

In the Zou et al. study, the melanoma lesions had an average diameter of 2.8 cm (0.5−4.6 cm).  What is more, the authors of those two studies reported enrolment of the ulcerated, crusted, bleeding, nodular, located on the heel, or melanomas with expansion on nail folds – providing evidence not only for an examination of clinically suspicious/evident acral malignant lesions but also commonly challenging to evaluate with Vivascope 1500 due to technical reasons. In clinical practice, the main application of the RCM is the differential diagnosis of equivocal melanocytic lesions presenting asymmetry of structures and/or colours and intermediate patterns under dermoscopy – and such studies are lacking.

Upon examination of the lesions on the palms and soles, it was found that skin scraping does not enable RCM to reach the DEJ, which provides only partial insight into the epidermis. Moreover, unreliable results might be obtained after considering the overlapping RCM structures between acral nevi and small early acral melanomas—therefore, its frequency should be determined. Those aspects were not pointed out in this article.

Based on this review, readers should gain more objective information (including pitfalls) on different aspects of RCM application in diagnosing acral melanocytic lesions.

The introduction should describe acral melanoma's benign, intermediate, and malignant dermoscopic patterns. 

Table 1 presents fault data regarding the reference of Cinotti et al., 2016 [29]—it should be 17 ALM and 26 acral nevi.

In the discussion, the authors presented the FL(LC)-OCT as a potentially more potent tool than RCM in this topic. So far, no publication has reported its application in acral melanocytic lesions; therefore, the FL(LC)-OCT description might be more concise.

Author Response

The authors provided a systematic review of the literature describing the application of RCM in diagnosing acral melanocytic lesions. Due to the lack of adequately designed studies in this field, only two retrospective and observational studies evaluated RCM structures/criteria for ALM. Still, only one of them included the comparator group of melanocytic nevi. Including the two case reports, this review describes 50 acral melanomas diagnosed with RCM.

The detailed analysis of the cited studies provided insights into the much more limited utility of the RCM in diagnosing ALM, as most cases included advanced and clinically evident melanomas, which wasn't pointed out in this article.

C: In the Zou et al. study, the melanoma lesions had an average diameter of 2.8 cm (0.5−4.6 cm).  What is more, the authors of those two studies reported enrolment of the ulcerated, crusted, bleeding, nodular, located on the heel, or melanomas with expansion on nail folds – providing evidence not only for an examination of clinically suspicious/evident acral malignant lesions but also commonly challenging to evaluate with Vivascope 1500 due to technical reasons. In clinical practice, the main application of the RCM is the differential diagnosis of equivocal melanocytic lesions presenting asymmetry of structures and/or colours and intermediate patterns under dermoscopy – and such studies are lacking.

R: Thanks for your suggestion. We highlighted such aspects in the limitations, as the existence of low data and the inclusion of clinically evident melamomas in the studies (Page 8 line 316: Indeed, only two research studies investigated RCM structures/criteria for ALM, and one of them made a comparison with melanocytic nevi. In addition, some melanomas were clinically evident in these studies, presenting as ulcerated, crusted, bleeding and nodular, or extending to the nail fold [29,34].)

 (C): Upon examination of the lesions on the palms and soles, it was found that skin scraping does not enable RCM to reach the DEJ, which provides only partial insight into the epidermis. Moreover, unreliable results might be obtained after considering the overlapping RCM structures between acral nevi and small early acral melanomas—therefore, its frequency should be determined. Those aspects were not pointed out in this article.

(R): In the case report of John et al [32], the authors of the article were able to obtain a deeper observation up to superficial dermis through skin debridement (a regular honeycomb pattern in the epidermis and dense dermal nests could be observed), so according to the manuscript this technique enables to examine epidermis, DEJ and superficial dermis.

(C): Based on this review, readers should gain more objective information (including pitfalls) on different aspects of RCM application in diagnosing acral melanocytic lesions.

R: We strongly agree with this comment, and we pointed out in the discussion the main limitations of RCM: technical limits (page 8 line 289: One of the main technical limitation is the loss resolution of RCM below 250µm in depth, that may influence the evaluation of skin lesions in acral sites. Such bias may be less relevant in the fingers or toes where the skin is thinner compared to other acral sites, and by mechanically removing the most superficial layer of the skin lesion examined [32]), and the scarse literature concerning this topic (page 8 line 314: Another significant limitation is the scarce literature describing RCM application on acral melanocytic lesions, mostly relying on case report/series and few original research studies. Indeed, only two research studies investigated RCM structures/criteria for ALM, and one of them made a comparison with melanocytic nevi. In addition, some melanomas were clinically evident in these studies, presenting as ulcerated, crusted, bleeding and nodular, or extending to the nail fold [29,34]. The low data and the heterogeneous in-vestigations explain why meta-analyses could not be conducted. Case reports and case series were however included as they provide relevant details that can contribute to tips in clinical practice)

 (C):The introduction should describe acral melanoma's benign, intermediate, and malignant dermoscopic patterns.

R: In the section introduction we have largely reported the main dermoscopic patterns of acral melanocityc nevi and ALM; also the different algorithms commonly used to detect ALM were listed. (Page 2 line 54: Dermoscopy has been widely used as an adjunctive clinical tool for the assessment of melanocytic and non-melanocytic acral skin lesions, and its application demonstrated to increase sensitivity and specificity for the diagnosis of ALM, by allowing the visualization of diagnostic clues not visible with naked eye [4,5]. Indeed, ALM shows specific dermoscopic patterns, including the parallel ridge pattern (PRP) and irregular diffuse pigmentation, in contrast to benign acral melanocytic nevi commonly identified by 1 of these major patterns: the parallel furrow pattern, the fibrillar pattern and the lattice-like pattern [4,5,8] (Figures 1, 2). With dermoscopy, a prediction of the tumor thickness (in situ versus invasive ALM) may be supposed before histopathologic exam, since in situ ALM reveals specific colors (blue, and white) and patterns (atypical vascular patterns, blue-white veil, and ulcers) with a reduced frequency rather than invasive form [9]. Two different dermoscopic algorithms, the 3-step algorithm and the BRAAFF algorithm, have been proposed for a more accurate) management of acquired melanocytic lesions on the acral sites [10,11]. The 3-step algorithm was introduced by Saida and Koga in 2007 and considered the presence of the parallel ridge pattern and diameter of skin lesion, to differentiate early ALM and acral nevi. Based on this algorithm, skin lesions showing parallel ridge pattern or measuring a diameter >7 mm without typical benign pattern, should be confirmed by histopathology; all other lesions may be clinically followed over time [10]. A revised version of the algorithm in 2011 suggested to remove clinical follow-up for acral nevi with benign dermoscopic findings, as ALM mostly arises de novo [12]. In a single retrospective chart review the 3-step algorithm was found to have a sensitivity of 80.0%, specificity of 87.8%, positive predictive value of 44.4%, and negative predictive value of 97.2% for the diagnosis of ALM [13]. Considering the low sensitivity of the parallel ridge pattern in ALM in their case series, Lallas et al proposed the BRAAFF checklist, to enhance the diagnosis of ALM presenting irregular blotch and asymmetry of structures and colours [11]. The scoring system evaluates 4 positive patterns: irregular blotches (1 point), parallel ridge pattern (3 points), asymmetry of structures (1 point), and asymmetry of colours (1 point); and 2 negative features: parallel furrow pattern (-1 point) and fibrillar pattern (-1 point). Lesions with total score of 1 or higher are suspicious for ALM and should be accurately investigated [11]. Parameters of sensitivity and specificity of the BRAAFF checklist were reported as 93.1% and 86.7%, respectively. [11] Although the use of dermatoscopy has improved the diagnostic accuracy for acral melanocytic lesions, differential may be challenging, and a missed diagnosis may be fatal [9-13].

(C): Table 1 presents fault data regarding the reference of Cinotti et al., 2016 [29]—it should be 17 ALM and 26 acral nevi.

(R): As suggested, data were corrected in Table 1

(C): In the discussion, the authors presented the FL(LC)-OCT as a potentially more potent tool than RCM in this topic. So far, no publication has reported its application in acral melanocytic lesions; therefore, the FL(LC)-OCT description might be more concise.

R: The lack of studies employing LC-OCT for acral lesions has been pointed out. Page 8, line 312: as currently this device has not been already employed to evaluate acral sites.

Reviewer 2 Report

Comments and Suggestions for Authors

Dear Authors,

I was pleased to read this interesting review. Melanoma is known for its aggressiveness that is why,  any paper that can highlight some potentials additional tools for diagnosis in this type of cancer, can open new horizons.

I think the authors should also add some dermoscopy images with the specific patterns they are talking about. The subject is about the analysis of some images and that is why I think it would be of great impact to  add some examples.

Clinicians who go through this review would appreciate this

Presenting the clinical value of a visual method without any image is a big minus for your journal and I am sure that the authors have such images in their own collection.

As reagarding drafting  the text:

Ø  bibliographic references are repeated quite often, for example, paragraphs 166-180 have only one reference

Ø  paragraps 189-197 only one reference

Ø  paragraphs 154-165 only one reference

Author Response

Dear Authors,

I was pleased to read this interesting review. Melanoma is known for its aggressiveness that is why,  any paper that can highlight some potentials additional tools for diagnosis in this type of cancer, can open new horizons.

(C): I think the authors should also add some dermoscopy images with the specific patterns they are talking about. The subject is about the analysis of some images and that is why I think it would be of great impact to  add some examples.

Clinicians who go through this review would appreciate this

Presenting the clinical value of a visual method without any image is a big minus for your journal and I am sure that the authors have such images in their own collection.

(R): Thanks for your suggestion. We added clinical and dermoscopic images of acral nevi (Figure 1, with different dermoscopic patterns) and of acral melanoma (Figure 2) in the introduction near the related description. The original Figure 1 was modified as Figure 3, because of the position in the text.

As reagarding drafting  the text:

Ø  bibliographic references are repeated quite often, for example, paragraphs 166-180 have only one reference

R: page 5 line 195: repeated reference was deleted

Ø  paragraps 189-197 only one reference

R: page 6, line 212: repeated reference was deleted

Ø  paragraphs 154-165 only one reference

R: page 6 line 218: repeated reference was deleted